# The Impact of Influencers on Cigar Promotions: A Content Analysis of Large Cigar and Swisher Sweets Videos on TikTok

**DOI:** 10.3390/ijerph19127064

**Published:** 2022-06-09

**Authors:** Jiaxi Wu, Alyssa F. Harlow, Derry Wijaya, Micah Berman, Emelia J. Benjamin, Ziming Xuan, Traci Hong, Jessica L. Fetterman

**Affiliations:** 1College of Communication, Boston University, Boston, MA 02215, USA; jiaxiw@bu.edu (J.W.); tjhong@bu.edu (T.H.); 2Department of Population and Public Health Sciences, Keck School of Medicine, University of Southern California, Los Angeles, CA 90032, USA; afharlow@usc.edu; 3Department of Computer Science, Boston University, Boston, MA 02215, USA; wijaya@bu.edu; 4College of Public Health & Moritz College of Law, The Ohio State University, Columbus, OH 43210, USA; berman.31@osu.edu; 5National Heart, Lung, and Blood Institute’s Framingham Heart Study, Framingham, MA 20892, USA; emelia@bu.edu; 6Section of Cardiovascular Medicine, Boston Medical Center, Department of Medicine, School of Medicine, Boston University, Boston, MA 02118, USA; 7Department of Epidemiology, School of Public Health, Boston University, Boston, MA 02118, USA; 8Department of Community Health Sciences, School of Public Health, Boston University, Boston, MA 02118, USA; zxuan@bu.edu; 9Evans Department of Medicine, Whitaker Cardiovascular Institute, School of Medicine, Boston University, Boston, MA 02118, USA

**Keywords:** cigars, little cigars, flavored cigars, Swisher Sweets, social media, influencer promotion, TikTok

## Abstract

Little is known about the content, promotions, and individuals in cigar-related videos on TikTok. TikTok videos with large cigar and Swisher Sweets-related hashtags between July 2016 and September 2020 were analyzed. Follower count was used to identify influencers. We compared content characteristics and demographics of featured individuals between cigar types, and by influencer status. We also examined the association between content characteristics and video engagement. Compared to large cigar videos, Swisher Sweets videos were more likely to feature arts and crafts with cigar packages, cannabis use, and flavored products. In addition, Swisher Sweets videos were also more likely to feature females, Black individuals, and younger individuals. Both Swisher Sweets and large cigar influencers posted more videos of cigar purchasing behaviors than non-influencers, which was associated with more video views. None of the videos disclosed sponsorship with #ad or #sponsored. Videos containing the use of cigar packages for arts and crafts, and flavored products highlight the importance of colorful packaging and flavors in the appeal of Swisher Sweets cigars, lending support for plain packaging requirements and the prohibition of flavors in cigar products to decrease the appeal of cigars. The presence and broad reach of cigar promotions on TikTok requires stricter enforcement of anti-tobacco promotion policies.

## 1. Introduction

In 2020, cigars were the most used combustible tobacco product with a higher prevalence of use than cigarettes among US youth [1]. Between 2009 and 2020, the increase in US cigar sales was primarily driven by the sale of flavored cigars [2]. Swisher Sweets, a leading flavored cigar brand, accounted for over 22% of market shares in US Convenience Stores in 2020 [3]. Factors that contribute to the increased cigar use among youth and young adults include the availability of flavors [4], small pack sizes [5], the industry’s targeted marketing [6], features that facilitate cannabis use [7], psychosocial factors [8], and reduced health risk perceptions of cigar smoking compared to cigarette smoking [9].

Cigars are broadly categorized into three types: large cigars, cigarillos, and little cigars [10]. The large cigar category includes both premium hand-rolled and machine-made large cigars. Cigarillos are short and narrow cigars that usually do not contain a filter and are available in a wide array of flavors. Little cigars are similar in size and shape to cigarettes and typically contain a filter. Historically, cigars were primarily premium cigars and were used mainly by older, predominantly white men who reported infrequent use and no inhalation [11]. As a result, cigars are not as heavily regulated and taxed as cigarettes [12].

Cigar companies promote product features prohibited in cigarettes, such as flavors and small pack sizes [13]. Regulatory loopholes have resulted in the generation of cigar products, specifically flavored little cigars and cigarillos (LCCs), designed to appeal to youth, young adults, and individuals of low-socioeconomic status [12]. As a result, the demographic characteristics of users, cigar product use patterns, purchasing behaviors, and reasons for use vary by cigar type [14]. Compared to users of premium large cigars, users of LCCs tend to be younger, non-Hispanic Black, have low educational attainment, and have low income. People who smoked LCCs are also more likely to report the use of a flavored, commonly used brand cigar than individuals who used premium cigars. Thus, it is critical to distinguish between people who use premium large cigars from people who use flavored LCCs, and to develop targeted public health and intervention efforts toward users of different cigar products [15].

Cigar smoking in the US also presents a critical health equity issue. Cigar companies use targeted strategies to promote products to communities of color [16]. As a result, the prevalence of cigar use is higher among non-Hispanic Black individuals than among other racial/ethnic groups [15]. In addition, Black youth are more likely to initiate tobacco use with cigars compared to White youth [17]. Recent surveillance data also indicate a higher prevalence of cigar use among high school students who are Black, compared to White and Hispanic youth [1].

Social media use is pervasive among youth, with 85% of youth using at least one social media site, and 45% say they are constantly online [18]. Social media has given rise to a class of non-traditional celebrities called influencers, who have large online followings and are valued as opinion leaders [19]. Influencers promoting tobacco products can potentially affect their followers’ attitudes and use of tobacco products. Followers of tobacco influencers are more likely to be an especially vulnerable group because they tend to be younger, have lower education, and are more likely to report past month tobacco use than those who do not follow tobacco influencers [20]. The positive association between exposure and engagement with tobacco-related social media content and tobacco use among youth warrants investigation into the prevalence of tobacco-related influencer promotions on social media [21].

Social learning theory provides a basis for explaining the effects of exposure to tobacco-related influencer posts on youth tobacco use [22]. Social learning theory posits that people acquire behaviors through observing, modeling, and imitating the behaviors of others [22]. Individuals who are observed are referred to as models. In real life, young people are surrounded by different types of influential models, from parents and teachers to peers. In contrast, on social media, models for behaviors are more limited, primarily to influencers who are often paid for sponsorships of products and services. Thus, by observing examples of behavior through social media, people, especially youth, are more likely to adopt the attitudes and behaviors exhibited by the influencer [23].

TikTok is the fastest-growing social media platform in the world and was the most downloaded mobile app in 2021 [24]. TikTok is especially popular among youth, with those 14 years and younger accounting for more than a third of TikTok’s 49 million daily users in the US [25]. On TikTok, users are continuously provided with new content, as videos start automatically one after another, unprompted [26]. On average, users spend 52 min on the platform and consume more than 200 videos per day, including carefully targeted ads [27]. TikTok’s user guidelines prohibit the posting of “content that depicts minors consuming, possessing, or suspected of consuming alcoholic beverages, drugs, or tobacco”, and the advertising or trade of tobacco products is also prohibited [28,29]; however, previous research found widespread tobacco promotions on Facebook, despite the platform’s policies prohibiting tobacco advertising [30]. It is unknown if TikTok’s anti-tobacco promotion policies are enforced, warranting further examination.

The goal of the current study was to examine the portrayals and promotions of large cigars and LCCs on TikTok, which is largely unexplored. Specifically, we compared the content features and individuals in videos of the different cigar types, and between influencer and non-influencer cigar videos. Our analysis of cigar-related videos on TikTok adds to the literature describing the differences in the users and use patterns of different types of cigar products. We also examined the association between content features and video popularity on TikTok.

## 2. Materials and Methods

### 2.1. Data Collection

We searched for different large cigar and LCC-related hashtags on TikTok to identify the most representative hashtags for cigar-related videos. Our initial observation on TikTok revealed that the hashtags “#cigar” and “#cigars” were commonly used in videos of traditional large cigars (see Appendix A for details on the number of videos identified by hashtag). We also searched hashtags #littlecigar, #littlecigars, #cigarillo, and #cigarillos to identify LCC-related videos; however, these LCC-related hashtags were rarely used by TikTok users. To identify content on TikTok related to LCCs, we extended our search to include Swisher Sweets-related hashtags (“#swisher”, “#swishers”, “#swishersweet”, “#swishersweets”), given that LCC users are more familiar with brand names rather than the terms “cigarillos” or “little cigars” [31] and are more likely to mention specific brands when posting about LCCs [32]. We found that Swisher Sweets-related hashtags were viewed over 16 million times on TikTok. Because Swisher Sweets is the leading cigar brand in the US for LCCs [33], is preferred among the US youth and young adults [34], and is commonly used as a keyword in previous social media research of LCCs [35,36,37,38], we only used Swisher Sweets-related hashtags to retrieve LCC-related TikTok videos for the current study.

TikTok’s Terms of Service [39] prohibits the use of public videos for commercial purposes, which was not the intent of this study. On September 17, 2020, using an open-source TikTok scraping tool [40], we scraped all 4361 publicly available videos with cigar and Swisher Sweets-related hashtags ever posted on TikTok. Pre-determined large cigar and Swisher Sweets hashtags were used to identify all publicly available TikTok videos that contained those hashtags up to the scraping date. We scraped: (1) 3456 videos with cigar-related hashtags and (2) 905 videos with Swisher Sweets-related hashtags. We also collected the associated metadata, including numbers of video views, likes, shares, and follower counts. Data for this study were stored in a password-protected computer and were only accessible to the authors. Research procedures were deemed to not meet the definition of human subjects research by the Authors’ Institutional Review Board due to the use of publicly available data.

### 2.2. Sample

Previous research has defined influencers as individuals with a minimum of 1000 followers [37]. For our study, we defined influencers as those with the top 75th percentile of total followers within each hashtag dataset because the follower counts varied between the large cigar and Swisher Sweets videos. Because people who smoke large cigars and Swisher Sweets differ in demographic features, cigar-smoking patterns, and purchasing behaviors [14], we sampled influencers for large cigar and Swisher Sweets videos separately; thus, instead of using a fixed number, which may cause biased sampling, we used the 75th percentile of followers within each hashtag to ensure we identified the top influential users for each of the two cigar types.

The minimum number of followers for an influencer was 15,000 and 1759 for large cigar videos and Swisher Sweets videos, respectively. We included all videos from influencers and randomly selected the matched numbers of videos from non-influencers for each of the respective hashtag categories. We also excluded non-English and non-relevant videos, resulting in a final sample of N = 1700 videos (1333 cigar videos and 367 Swisher Sweets videos, Figure 1).

### 2.3. Video Coding and Inter-Coder Reliability 

We created a coding scheme of fifteen content features identified from previous studies and that emerged from the current dataset. Specifically, we identified nine themes from previous social media analyses of cigar and LCC-related posts including: (1) Product promotion [35]; (2) Smoking cigars [32,35]; (3) Cannabis use (e.g., removing some, or all the tobacco from the cigar and replacing it with cannabis, Figure 2A) [35]; (4) Smoke trick [35]; (5) Prevention [32]; (6) Flavor [38]; (7) Purchasing behaviors (Figure 2B) [38]; (8) Product review (Figure 2C) [41]; and (9) Cigar-related marketing events (Figure 2D) [36]. We further identified six themes observed from the current dataset including: (10) Arts and crafts with cigar packages (Figure 2E); (11) Individuals dancing with background music referring to cigar smoking (making smoking gestures); (12) Use of the Cardi B Swisher Sweets song as video background music; (13) Showing off multiple Swisher Sweets packages (Figure 2F). We additionally coded: (14) Written health warnings; and (15) Audio health warnings based upon previous research suggesting that health warnings in tobacco-related social media posts results in a more negative tobacco brand perception [42]. Table 1 displays descriptions of the coded video content features. Video content features were not mutually exclusive, meaning that a video could contain multiple content features simultaneously.

For videos containing people, we also coded for the following demographic features: (1) perceived sex—the presence of males and females; (2) perceived race—the presence of Black, White, Asian, and Hispanic or Latino individuals; and (3) perceived age—the presence of younger or older individuals. Coders used all available visual and audial cues (e.g., skin color, background voices, appearances) to inform their coding choices. Consistent with a previous study that coded social media users’ age from their profile pictures [32], we assigned “younger” to individuals who look like they were under the age of 21 (i.e., individuals who look like teens in middle or high school to young adults under the age of 21) and older individuals (≥21 years of age) based on visual and audio cues in the videos. In the event demographic features were difficult to determine, coders could select “unknown” for the sex/race/age of featured individuals if none of the visual and audial cues were available to determine the demographics.

To attain high coding reliability, three coders were first trained on 20 videos together to ensure the visual/audio cues used to code the content features and demographic categories were consistent across all coders. Next, three coders independently coded 50 videos, after which discrepancies were discussed to resolve coding disagreements. Coders first determined whether the video was in English and relevant to cigars and Swisher Sweets. A total of 280 videos (273 large cigar and 7 Swisher Sweets videos) were not in the English language and were excluded from the analyses. A video was considered relevant only when the video showed or referred to cigar smoking. An example of non-relevant videos included videos of Swisher brand lawn mowers. Next, following the coding scheme, coders identified the content features and demographics of individuals in the videos. The inter-coder reliability was calculated using 10% (N = 221) of the sample. Coding agreements were assessed with Cohen’s Kappa values, which were above 0.7 across all content variables, indicating a high level of intercoder reliability [43]. Three coders independently coded the remaining videos in the sample. 

### 2.4. Statistical Analyses

We performed chi-square tests to compare the content features and individual characteristics for large cigar and Swisher Sweets videos, between influencer’s and non-influencer’s videos within each of the cigar types, and between large cigars and Swisher Sweets influencers’ videos. When comparing large cigars and Swisher Sweets videos, we excluded the content features (1) use of the Cardi B Swisher Sweets song as video background music” and (2) showing off multiple Swisher Sweets packages from the analysis because these features were unique to Swisher Sweets videos. We additionally calculated odds ratios and 95% confidence intervals. Chi-square analyses were conducted using SPSS (Version 26) with an alpha level of 0.05 (2-tailed) with Bonferroni correction to account for multiple testing.

### 2.5. Modeling Video Engagement with Video Content Features

To identify the video content features associated with engagement (i.e., number of views, likes, and shares) of a large cigar or Swisher Sweets videos, we formulated negative binomial models for views and likes and negative binomial hurdle models for shares within each of the large cigar and Swisher Sweets datasets (see *SI* Section 2 for modeling details).

For each of the main models predicting video popularity (i.e., views, likes, shares), we used video content features as predictors. Given that some content features were rare in the data sets, we only included content features that appeared more than ten times within each dataset. We also adjusted for follower counts, which can potentially affect the engagement of social media posts, including videos on TikTok. Negative binomial and hurdle models were fitted using the glmmTMB package in R (version 4.1.0). A *p*-value less than 0.05 (two-tailed) was considered statistically significant. The *p*-values of each negative binomial and hurdle model were adjusted with the Bonferroni correction method.

### 2.6. Hashtag Analysis of Video Descriptions

To prohibit misleading or deceptive advertising, the Federal Trade Commission (FTC) requires that any type of online sponsored content must clearly disclose sponsorship [44]. For each of the video post descriptions, we analyzed whether the video description contained the FTC recommended sponsorship disclosure hashtags #ad and #sponsored [44]. The FTC requires disclosure of any financial relationship to the brand (including the provision of free products) and suggests the use of hashtags as one method of disclosure [45]. We used string matching techniques in R (Version 4.1.0) to determine if the description of a post for a video contained either of the two FTC recommended hashtags #ad and #sponsored.

## 3. Results

At the time of the study, the 1700 videos with large cigar and Swisher Sweets-related hashtags had been viewed over 159 million times on TikTok. The median follower counts for large cigar and Swisher Sweets hashtag videos were 14,900 and 1740, respectively. With respect to video engagement, influencers’ cigar videos received an average of 88,749 views, 4455 likes, and 41 shares; non-influencers’ cigar videos attracted an average of 8718 views, 304 likes, and 7 shares. Influencers’ Swisher Sweets videos received an average of 45,625 views, 4962 likes, and 116 shares; non-influencers’ Swisher Sweets videos elicited an average of 2434 views, 150 likes, and 10 shares (See Appendix A for details of video engagement for the two cigar types). For large cigar videos, the top two prevalent video themes were smoking cigars (59.2%) and product review (13.5%). For Swisher Sweets videos, the top two themes were flavor (48.8%) and cannabis use (28.9%) (See Appendix A). Coders also determined if Swisher Sweets LCCs or packaging were included in large cigar videos. Coders found that none of the 1333 sampled large cigar videos contained a Swisher Sweets LCC product or packaging, which is consistent with previous research that suggests users of LLC often mentioned specific brands when posting about LCCs, instead of using general tobacco product terms such as “cigar” and “cigars” [32].

### 3.1. Comparisons of Video Content Features and Featured Individual Demographics

Chi-square analyses comparing large cigar and Swisher Sweets videos indicated that large cigar videos contained more product promotions (*p* = 0.005), product reviews (*p* < 0.001), and individuals smoking cigars (*p* < 0.001) compared to Swisher Sweets videos. Swisher Sweets videos contained more videos of purchasing behaviors (*p* < 0.001), arts and crafts with cigar packages (*p* < 0.001), cannabis use (*p* < 0.001), dancing individuals making smoking gestures (*p* < 0.001), and flavored products (*p* < 0.001) compared to large cigar videos (See Appendix A). Large cigar videos were more likely to feature males (*p* < 0.001), while Swisher Sweets videos were more likely to show females (*p* < 0.001). In addition, large cigar videos contained more White individuals than Swisher Sweets videos (*p* < 0.001). In contrast, Swisher Sweets videos were associated with Black (*p* = 0.017), Asian (*p* = 0.001), and younger individuals (*p* < 0.001) (See Appendix A).

Compared to large cigar non-influencers, large cigar influencers were more likely to post about purchasing behaviors (*p* = 0.025) and product reviews (*p* < 0.001). For Swisher Sweets videos, influencers were also more likely to post purchasing behaviors content (*p* < 0.001), dancing individuals (*p* < 0.001), and flavored products (*p* = 0.041) compared to Swisher Sweets non-influencers (See Appendix A). Interestingly, large cigar influencers were less likely to be younger compared to the non-influencers (*p* = 0.009), while the opposite association was observed for Swisher Sweets influencers, who were more likely to be younger than non-influencers (*p* = 0.002) (See Appendix A).

Lastly, Chi-square analyses comparing large cigar influencer posts and Swisher Sweets influencer posts suggested that large cigar influencers were more likely to post product promotions (*p* = 0.020), product reviews (*p* < 0.001), and smoking individuals (*p* < 0.001); however, Swisher Sweets influencers were more likely to post videos of purchasing behaviors (*p* < 0.001), arts and crafts with cigar packages (*p* < 0.001), cannabis use (*p* < 0.001), dancing individuals making smoking gestures (*p* < 0.001), and flavored products (*p* < 0.001) (See Appendix A). Lastly, we found large cigar influencers were more likely to be males (*p* < 0.001) and White individuals than Swisher Sweets influencers (*p* < 0.001). On the contrary, when compared to large cigar influencers, Swisher Sweets influencers were more likely to be females (*p* < 0.001), Asian (*p* < 0.001), and young individuals (*p* < 0.001) (See Appendix A).

### 3.2. Predicting Video Popularity with Video Content Features

When interpreting results of a predictor from negative binomial and Hurdle models, all other predictors are held constant. For large cigar videos, negative binomial models showed that after adjusting for the number of account followers and other content feature predictors, video content of purchasing behaviors (IRR = 29.03, *p* < 0.001, CI = 17.03, 49.47) and product reviews (IRR = 2.50, *p* < 0.001, CI = 1.98, 3.17) elicited more likes. Purchasing behaviors (IRR = 20.92, *p* < 0.001, CI = 11.98, 36.53) and product reviews (IRR = 2.03, *p* < 0.001, CI = 1.58, 2.61) also had a greater number of views compared to videos without these content features. We found that product promotions (IRR = 0.30, *p* < 0.001, CI = 0.21, 0.44) and content of an individual smoking cigars (IRR = 0.65, *p* < 0.001, CI = 0.54, 0.77) was associated with fewer likes. Product promotions (IRR = 0.32, *p* < 0.001, CI = 0.22, 0.47) and content of individuals smoking cigars (IRR = 0.56, *p* < 0.001, CI = 0.46, 0.67) were also associated with fewer views (See Appendix A). As for shares, the zero portion of Hurdle models revealed no significant results for any of the content features. The positive portion of Hurdle model suggested that videos of purchasing behaviors (IRR = 44.70, *p* < 0.001, CI = 9.13, 218.86) predicted more shares. Consistent with likes and views, product promotions (IRR = 0.29, *p* = 0.019, CI = 0.12, 0.74) and content of individuals smoking cigars (IRR = 0.47, *p* = 0.007, CI = 0.29, 0.77) were associated with fewer shares (See Appendix A). 

For Swisher Sweets videos, negative binomial analyses showed that, after adjusting for the number of account followers and other content feature predictors, videos that contained content on purchasing behavior received more video views (IRR = 2.96, *p* = 0.036, CI = 1.26, 7.00); however, video content of an individual smoking cigars was associated with fewer video likes (IRR = 0.37, *p* = 0.005, CI = 0.21, 0.68) and fewer video views (IRR = 0.32, *p* = 0.001, CI = 0.18, 0.59) (See Appendix A). When predicting video shares, none of the content features were associated with video shares in the zero portion of the model. The non-zero portion Hurdle model showed that smoking cigars also led to fewer shares of a Swisher Sweets post on TikTok (IRR = 0.01, *p* = 0.008, CI = 0.00, 0.17) (See Appendix A).

### 3.3. Disclosure of Sponsorship in Video Description

Text analyses of all of the 1700 cigars and Swisher Sweets video descriptions showed that none of the video descriptions contained hashtags that disclosed sponsorship, including #ad and #sponsored. Only three large cigar videos contained oral health warnings or disclaimers.

## 4. Discussion

The demographic characteristics, including age, sex, and race/ethnicity, differ between users of large cigars and LCCs [14]. Research suggests that socially disadvantaged communities, especially non-Hispanic Black individuals, are more likely to smoke cigars and to develop established cigar-smoking behaviors compared with non-Hispanic White individuals [46]. Our findings confirmed the literature on the demographic differences between users posting about large cigars or Swisher Sweets on TikTok. We found that Swisher Sweets TikTok videos were more likely to feature females, younger, Black, and Asian individuals compared to large cigar TikTok videos. When comparing between influencers and non-influencers’ videos, we found that compared to large cigar influencers, Swisher Sweets influencers tend to be younger and were more likely to be female. Future prevention campaigns for cigar products (i.e., large cigars and LCCs) should consider the different use patterns and demographics of users.

Swisher Sweets is the leading LCC brand popular among youth and young adults in the US [3]. Our study sheds light on the Swisher Sweets-related content that youth and young adults may be exposed to on TikTok, the fastest growing social media platform in the world that appeals to a younger population [24,25]. Future studies are needed to evaluate the content of videos of additional cigar brands to gain a more comprehensive understanding of LCC influencer promotions on TikTok.

Consistent with previous research across other social media platforms [35,37], our study noted that Swisher Sweets videos on TikTok were more likely to feature cannabis use and flavored products compared to videos featuring large cigars [35,38]. In addition, our study found that both large cigar and Swisher Sweets influencers were more likely to post about purchasing behaviors than non-influencer users. Large cigar influencers also posted product review videos more frequently than non-influencer users—a common promotional strategy of large cigars observed in traditional media such as magazines [47]. Our findings suggest that content of purchasing behavior and product reviews lead to more views of large cigar and Swisher Sweets videos. Future research is needed to investigate how exposure to short-form videos of cigar-related content on TikTok affect youth’s attitudes towards cigars and their susceptibilities to initiate and use cigars.

Exposure to celebrity-endorsed LCC promotions correlates with brand-specific cigar smoking among young adult smokers [48]. We found that one out of ten Swisher Sweets videos on TikTok used Cardi B’s “Swisher Sweets” song as the background music, suggesting that celebrity celebration of tobacco products can echo widely on social media. Moreover, our study also revealed that TikTok users, through arts and crafts, are re-purposing Swisher Sweets packages to streetwear and paraphernalia emblazoned with the Swisher Sweets logo. Many LCC brands, including Swisher Sweets, use colorful and shiny packages that boldly communicate the flavor appeal to youth and young adults. Visual cues from the cigar packaging impact young adults’ affect and increase their susceptibility to cigar smoking [49]. Regulations on the packaging of flavored cigars, including the use of pictorial warning labels, plain packaging, and restrictions on celebrity endorsement and other youth appealing promotional content may decrease the appeal and use of cigar products among youth.

To prohibit misleading or deceptive advertising, the FTC requires that any type of online sponsored content must clearly disclose sponsorship. For sponsored social media posts, the FTC recommends the use of clear hashtags such as “#ad” and “#sponsored”, or other similarly clear methods to disclose sponsorship [44]. None of the videos in our analyzed data set contained hashtags indicating sponsorship. We acknowledge that we do not have access to data regarding a financial transaction between the manufacturers and the influencers; however, given that many of the influencers in our dataset were retailers and cigar manufacturers, the promotional regulations could also be applied to those users. Future research may examine the prevalence and potential effects of exposure to promotions from retailers and cigar manufacturers on purchasing intentions and behaviors. Research has indicated that labeling commercially sponsored tobacco content on social media with #ad and #sponsored attracts the attention of youth and young adults, making it a viable strategy to inform audiences of the promotional nature of the posts [50]. Sponsorship disclosure promotes critical evaluation of the promotional post and ultimately decreases advertising effectiveness [51]. A study found that clear sponsorship disclosure decreased young adult participants’ perceptions of influencer credibility and their intentions to engage with e-cigarette Instagram posts [52]. Examining the effects of influencer posts and regulatory interventions in influencer promotions on youth tobacco experimentation is an important direction for future research.

Even though TikTok prohibits the promotion of tobacco products and the posting of minors consuming tobacco products [29], we still observed such content on TikTok. Exposure and engagement with tobacco-related social media content are associated with tobacco use among youth [21]. Because more than a third of TikTok’s daily users are under the age of 14 [25], it is crucial to restrict the promotion of youth-appealing tobacco content on TikTok to reduce the effects of such promotions on tobacco use among youth.

Similar to a study that found that few, if any, influencers’ cigar-related posts on Twitter contained health warnings [37], we observed that only three of all large cigar and Swisher Sweets promotional videos contained audio health warnings or disclaimers, and no written health warning labels were observed. Prior research has reported that the inclusion of health warning statements in tobacco-related social media posts results in more negative brand perceptions [52]; therefore, from a public health perspective, it is important to enforce disclosures of sponsorship and health warnings on all content promoting tobacco products and to remove any content featuring the use of tobacco by underage users.

Our study has several limitations. Our study was cross-sectional and observational and hence, we cannot rule out residual confounding or establish causation. We also cannot exclude misclassification of the features we coded. For instance, identification of an individual’s demographic features was by external appearance and auditory cues, which may not be as accurate as self-reported demographic data, especially for age; however, we attempted to limit misclassification by including the option of coders to select unknown for the demographic identification. In addition, coders were instructed to code the individuals as over 21-years old when they were in doubt about the age determination. It is possible that we may be under-estimating the number of videos featuring younger individuals. We also do not know the tobacco use status of the individuals who engaged with the content; thus, we cannot determine causation or whether engagement with influencers’ cigar videos leads to cigar experimentation.

In violation of FTC guidance [44], many influencers do not use the recommended methods of disclosure, such as the use of the hashtags #ad or #sponsored to indicate that they have a “material connection” with a brand. As a result, paid influencer posts can be difficult to identify and study. We identified influencers as those in the top 75th percentile for the number of followers but acknowledge that we may be missing influencers who do not fall within the 75th percentile or include individuals with large numbers of followers who have no material connections to the industry. Previous research has identified LCC influencers with follower counts by categorizing users with 1000 and more followers as influencers [10]. Our study utilized a novel approach that considers the distribution of follower counts; however, the selection of the 75th percentile was arbitrary. To address this limitation, we conducted a sensitivity analysis identifying influencers as those individuals in the top 90th percentile for followers (Appendix A) and compared the results against those within the top 75th percentile. The sensitivity analysis resulted in similar findings when comparing the individual demographics and content features in videos of the two cigar types (see Appendix A for details on the sensitivity analysis), lending validation to our methodlogy. Our study focused on the TikTok platform and a single brand, Swisher Sweets; hence, our findings may not be generalizable to other social media platforms or other LCC brands. We studied the English language content on TikTok; the generalizability to other languages is unknown. Despite these limitations, the content and user characteristics identified in this study could inform the design of media campaigns and the development of tobacco control efforts.

## 5. Conclusions

In summary, our study found that the demographics of the featured individuals and content features differ between videos of large cigar and Swisher Sweets on TikTok. Specifically, compared to large cigar videos, Swisher Sweets videos were more likely to contain content of arts and crafts with cigar packages, cannabis use, and flavored products. Swisher Sweets videos were also more likely to feature individuals who are female, younger, Black, and Asian compared to large cigar TikTok videos. In addition, Swisher Sweets influencers’ videos tend to feature younger, and female individuals than large cigar influencers’ videos. We also identified specific content features that may facilitate the engagement of large cigar and Swisher Sweets videos on TikTok. Both Swisher Sweets and large cigar influencers posted more videos of cigar purchasing behaviors than non-influencers, which was associated with more video views. For large cigar videos, the content of product reviews was associated with more video views and likes on TikTok.

Our findings may inform the design and implementation of cigar prevention campaigns with targeted demographic populations. Our findings lend support for enforcement of disclosures of sponsorship and health warnings on TikTok and related regulatory actions to restrict the promotions of youth-appealing tobacco content on social media. Our findings also suggest an urgent need for the prohibition of flavors in cigars and extension of plain packaging to cigar products.

## Figures and Tables

**Figure 1 ijerph-19-07064-f001:**
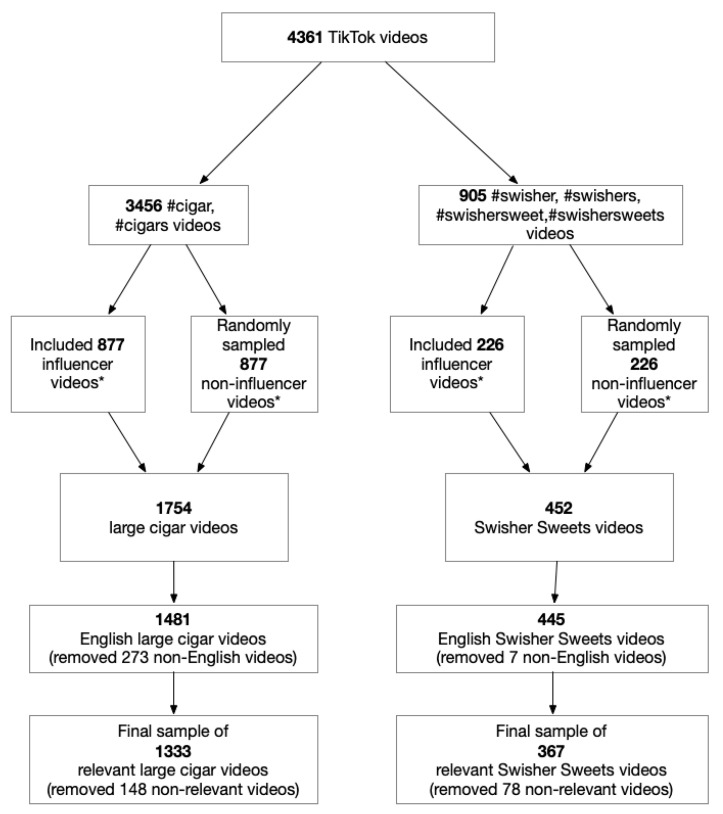
Data sampling procedure. Influencers’ videos were identified by defining influencers as those individuals with the top 75th quantile of number of followers within each hashtag category. The same number of non-influencers’ videos were randomly sampled for each hashtag category. The final sample of large cigar and Swisher Sweets videos was N = 1700 after excluding non-English videos.

**Figure 2 ijerph-19-07064-f002:**
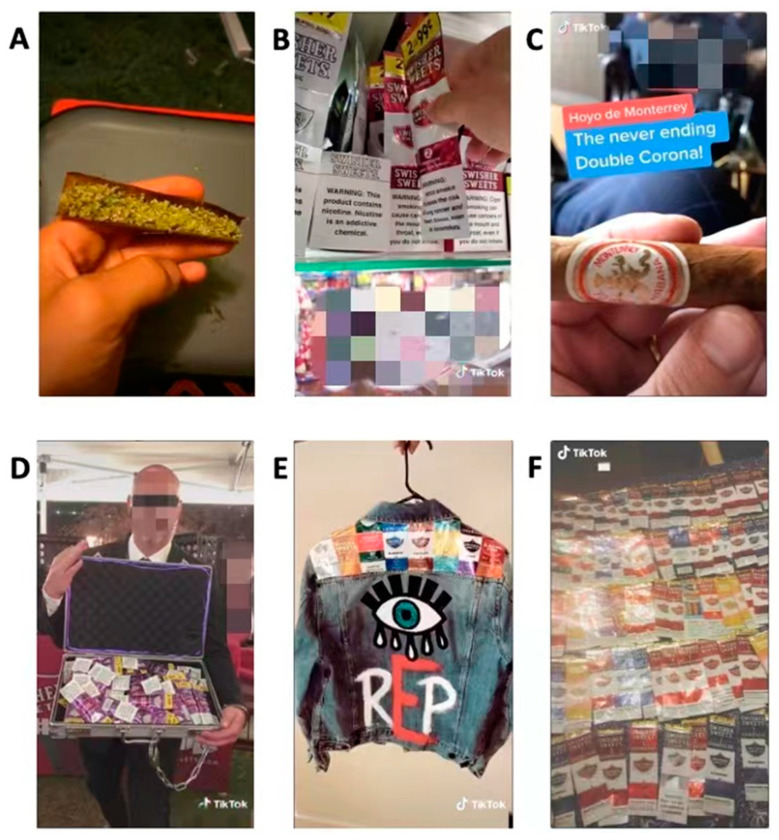
Representative images of select video themes. (**A**) Cannabis use: a cigar that has been hollowed out and filled with cannabis; (**B**) Purchasing behavior: a video of a user purchasing cigar products in a convenience store; (**C**) Product review: a video review of the “double corona” large cigar; (**D**) Cigar-related marketing events: a video of a Swisher Sweets marketing event; (**E**) Arts and crafts: a video of paraphernalia emblazoned with the Swisher Sweets logo through arts and crafts; (**F**) Showing off multiple Swisher Sweets packages: a video of one individual showing off all of the Swisher Sweets packages the person has smoked.

**Table 1 ijerph-19-07064-t001:** Descriptions of content features in English and relevant cigar/Swisher Sweets videos.

	Descriptions
**Video Content Features**	
Product Promotion	A video selling cigar products or promoting cigar stores, and professional ads
Smoking Cigars	A video showing individuals smoking featured cigar products
Cannabis Use	A video of cannabis use (e.g., blunt: a cigar that has been hollowed out and filled with cannabis)
Smoke Trick	A video of smoke tricks such as making smoke rings
Prevention	A video with a main theme of the negative effects and prevention of cigar and LCC products
Flavor	A video showing or referring to flavored cigar products
Purchasing Behavior	A video of accessing and purchasing cigar or LCC products
Product Review	A video commenting on or reviewing flavors, tastes, or features of cigar or LCC products
Cigar-related Marketing Events	A video promoting cigar companies’ marketing events (e.g., musical events)
Arts and Crafts with Cigar Packages	A video of using cigar/LCC products’ packages to create arts and crafts such as a rolling tray
Individual Dancing	A video of people dancing and making smoking gestures without smoking real cigar products
Cardi B Swisher Sweets Music	A video using Cardi B’s “Swisher music” as background music
Showing off multiple Swisher Sweets packages	A video showing more than five Swisher Sweets packages simultaneously
Written Health Warnings	A video containing written form of health warnings or disclaimers of cigar/LCC smoking superimposed on the video
Audio Health Warnings	A video containing oral form of health warnings or disclaimers of cigar/LCC smoking
**Sex**	
Presence of Males	A video featuring males
Presence of Females	A video featuring females
**Race**	
Black	A video featuring Black individuals
White	A video featuring White individuals
Spanish/Hispanic	A video featuring Spanish/Hispanic individuals
Asian	A video featuring Asian individuals
**Presence of Young Individuals**	A video featuring individuals who look like teens in middle/high school to people who are under the age of 21

## Data Availability

The code and data underlying this article will be shared on reasonable request to the corresponding author.

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
