# Peer review of "The Impact of Influencers on Cigar Promotions: A Content Analysis of Large Cigar and Swisher Sweets Videos on TikTok"

_ijerph, 2022, doi:10.3390/ijerph19127064_

Round 1
Reviewer 1 Report
The manuscript presents relevant content for scientific publication. However, the manuscript should be developed within the theoretical framework and critical thinking. It is essential to deepen the literature review and add references (Scopus and/or WoS) from the last 3-4 years. The manuscript has little scientific reflection and should develop contributions to theory, management and practice. It is suggested that limitations and future research should be added. English should be reviewed moderately. Introduced figures should include higher quality and greater sharpness. The research topic is relevant and current.
Author Response
We appreciate the constructive feedback from the Reviewers. We have worked to address and integrate the feedback of each reviewer in the revised, resubmitted manuscript. Specifically, in response to Reviewer 1, we have provided additional background in the Introduction explaining the rationale for comparing large cigar- and LCC-related videos on TikTok, the implications of our findings, additional discussion on the underlying theory of the study design, and up-dated the references with more recent data. Please see our detailed responses to Reviewer 1 below:
- The manuscript should be developed within the theoretical framework and critical thinking. It is essential to deepen the literature review and add references (Scopus and/or WoS) from the last 3-4 years.The manuscript has little scientific reflection and should develop contributions to theory, management and practice.
Response: We added additional references and content on: 1) the rationale for the comparison of large cigar- and LCC-related videos, 2) potential effects of social media and influencer promotions on youth tobacco use, 3) discussion of Albert Bandura’s social learning theory that underscored the design of the current study, and 4) what is unique about TikTok and why studying tobacco-related content on TikTok is significant.
Manuscript changes: In lines 49-67, we explain the rationale for comparing the content of large cigars and LCC-related videos: “Cigars are broadly categorized into three types: large cigars, cigarillos, and little cigars.1 The large cigar category includes both premium hand-rolled and machine-made large cigars. Cigarillos are short and narrow cigars that usually do not contain a filter and are available in a wide array of flavors. Little cigars are similar in size and shape as cigarettes and typically contain a filter. Historically, cigars were primarily premium cigars and were used mainly by older, predominantly white men who reported infrequent use and no inhalation.2 As a result, cigars are not as heavily regulated and taxed as cigarettes.3
Cigar companies promote product features prohibited in cigarettes, such as flavors and small pack sizes.4 Regulatory loopholes in cigars have resulted in the generation of cigar products, specifically flavored cigarillos and little cigars (LCCs), designed to appeal to youth, young adults, and individuals of low-socioeconomic status.3 5 6 As a result, the demographic characteristics, cigar product use patterns, purchasing behaviors, and reasons for use vary by cigar types.7 Compared to users of premium large cigars, users of LCCs tend to be younger, non-Hispanic black, have low educational attainment, and have low income. People who smoked LCCs are also more likely to report the use of a flavored usual brand cigar than individuals who used premium cigars. Thus, it is critical to distinguish between people who use premium large cigars and flavored LCCs, and to develop targeted public health and intervention efforts toward users of different cigar products.8”
In lines 75-84, we added content on the potential effects of social media and influencer promotions on youth tobacco use: “Social media use is pervasive among youth, with 85% of youth using at least one social media site, and 45% say they are constantly online.9 Social media has given rise to a class of non-traditional celebrities called influencers, who have large online followings and a high value as opinion leaders.10 Influencers discussing tobacco use can potentially affect their followers’ attitudes and use of tobacco products. Followers of tobacco influencers are more likely to be an especially vulnerable group who are younger, have lower education, and are more likely to report past month tobacco use than those who do not follow tobacco influencers.11 The positive association of exposure and engagement with tobacco-related social media content and tobacco use among youth warrants investigation into the prevalence of tobacco-related influencer promotions on social media.12”
In lines 85-93 we added content on the social learning theory, which underscores the current study: “Social learning theory provides a basis for explaining the effects of exposure to tobacco-related influencer posts on youth tobacco use.13 Social learning theory emphasizes the importance of observing, modeling, and imitating the behaviors of others. New patterns of behavior can be acquired through observation of the behaviors of others.13 Individuals who are observed are referred to as models. In real life, young people are surrounded by different types of influential models, such as parents and teachers. On social media, influencers serve as models. By observing examples of behavior through social media, people, especially youth, are more likely to adopt the attitudes and behaviors exhibited by the influencer.14”
In lines 94-105, we added content on why we studied cigar-related content on TikTok: “TikTok is the fastest growing social media platform in the world and was the most downloaded mobile app in 2021.15 TikTok is especially popular among youth. More than a third of TikTok’s 49 million daily users in the US are aged 14 or younger16. On TikTok, users are continuously provided with new content, as videos start automatically one after another, unprompted.17 On average, users spend 52 minutes on the platform and consume more than 200 videos per day, including carefully targeted ads.18 TikTok’s user guidelines prohibit the posting of “content that depicts minors consuming, possessing, or suspected of consuming alcoholic beverages, drugs, or tobacco,” and the advertising or trade of tobacco products is also prohibited.19 20 However, previous research found widespread tobacco promotions on Facebook, despite the platform’s policies prohibiting tobacco advertising.21 It is unknown if TikTok’s anti-tobacco promotion policies are enforced, warranting further examination.”
- It is suggested that limitations and future research should be added.
Response: We discuss future research directions in lines 343-345, 349-351, 360-362, 384-386, 392-394, and the last two paragraphs (lines 409-432) contain our study limitations.
Manuscript changes: We discuss future directions after the discussion of each major finding in the discussion section, which are located at:
Lines: 343-345: “Future prevention campaigns of cigar products (i.e., large cigars and LCCs) should consider the different use patterns and demographics of users.”
Lines: 349-351: “Future studies are needed to evaluate the content of videos of additional cigar brands to gain a more comprehensive understanding of LCC influencer promotions on TikTok.”
Lines: 360-362: “Future research is needed to investigate how exposure to short-form videos of cigar-related content on TikTok affect youth’s attitudes towards cigars and their susceptibilities to initiate and use cigars.”
Lines: 384-386: “Future research may examine the prevalence and potential effects of exposure to promotions from retailors and cigar manufacturers on purchasing intentions and behaviors.”
Lines: 392-394: “Examining the effects of influencer posts and regulatory interventions in influencer promotions on youth tobacco experimentation is an important direction for future research. ”
The limitations of our study can be found in the last two paragraphs (lines 409-432):
“Our study has several limitations. Our study was cross-sectional and observational and hence, we cannot rule out residual confounding or establish causation. We also cannot exclude misclassification of the features we coded. For instance, identification of an individual’s demographic features was by external appearance and auditory cues, which may not be as accurate as self-reported demographic data, especially for age. However, we attempted to limit misclassification by including the option of coders to select unknown for the demographic identification. In addition, coders were instructed to code the individuals as over 21-years old when they were in doubt in any way of the age determination. It is possible that we may be under-estimating the number of videos featuring younger individuals. We also do not know the tobacco use status of the individuals who engaged with the content; thus, we cannot determine causation or whether engagement with influencers’ cigar videos leads to cigar experimentation.
In violation of Federal Trade Commission guidance,22 many influencers do not use the recommended methods of disclosure such as the use of the hashtags #ad or #sponsored to indicate that they have a “material connection” with a brand. As a result, paid influencer posts can be difficult to identify and study. We identified influencers as those in the 75th percentile for the number of followers but acknowledge that we may be missing influencers who do not fall within the top 75th percentile, or including individuals with large numbers of followers who have no material connections to the industry. Our study focused on the TikTok platform and a single brand, Swisher Sweets; hence, our findings may not be generalizable to other social media platforms or other LCC brands. We studied English language TikTok; the generalizability to other languages is unknown. Despite these limitations, the content and user characteristics identified in this study could inform the design of media campaigns and the development of tobacco control efforts.”
- Introduced figures should include higher quality and greater sharpness.
Response: We replaced Figure 1 with a higher quality image and increased the sharpness of the image. However, Figure 2 contains screenshots of TikTok videos, which are by nature of the format, of low quality.
References
- CDC. Cigars 2022 [Available from: https://www.cdc.gov/tobacco/data_statistics/fact_sheets/tobacco_industry/cigars/index.htm accessed May 16 2022.
- National Cancer Institute. Cigars: Health Effects and Trends Tobacco Control Monograph No 9 Bethesda, MD: US Department of Health and Human Services, National Institutes of Health, National Cancer Institute NIH Pub (No. 98-4302)
- Delnevo CD, Hrywna M, Giovenco DP, et al. Close, but no cigar: certain cigars are pseudo-cigarettes designed to evade regulation. Tob Control 2017;26(3):349-54. doi: 10.1136/tobaccocontrol-2016-052935 [published Online First: 2016/05/26]
- Delnevo CD, Giovenco DP, Miller Lo EJ. Changes in the Mass-merchandise Cigar Market since the Tobacco Control Act. Tob Regul Sci 2017;3(2 Suppl 1):S8-s16. doi: 10.18001/trs.3.2(suppl1).2 [published Online First: 2017/03/21]
- National Center for Chronic Disease Prevention and Health Promotion (US) Office on Smoking and Health. Preventing Tobacco Use Among Youth and Young Adults: A Report of the Surgeon General. Atlanta (GA): Centers for Disease Control and Prevention (US); 2012. [Available from: https://www.ncbi.nlm.nih.gov/books/NBK99237/ accessed July 3 2021.
- Sterling KL, Fryer CS, Nix M, et al. Appeal and Impact of Characterizing Flavors on Young Adult Small Cigar Use. Tobacco regulatory science 2015;1:42-53. doi: 10.18001/TRS.1.1.5 [published Online First: 2015/03/01]
- Corey CG, Holder-Hayes E, Nguyen AB, et al. US Adult Cigar Smoking Patterns, Purchasing Behaviors, and Reasons for Use According to Cigar Type: Findings From the Population Assessment of Tobacco and Health (PATH) Study, 2013-2014. Nicotine Tob Res 2018;20(12):1457-66. doi: 10.1093/ntr/ntx209 [published Online First: 2017/10/24]
- Weinberger AH, Delnevo CD, Zhu J, et al. Trends in Cigar Use in the United States, 2002–2016: Diverging Trends by Race/Ethnicity. Nicotine & Tobacco Research 2020;22(4):583-87. doi: 10.1093/ntr/ntz060
- Pew Research Center. Teens, Social Media and Technology 2018 2018 [Available from: https://www.pewresearch.org/internet/2018/05/31/teens-social-media-technology-2018/ accessed July 3 2021.
- De Veirman M, Cauberghe V, Hudders L. Marketing through Instagram influencers: the impact of number of followers and product divergence on brand attitude. International Journal of Advertising 2017;36(5):798-828. doi: 10.1080/02650487.2017.1348035
- Chu KH, Majmundar A, Allem JP, et al. Tobacco Use Behaviors, Attitudes, and Demographic Characteristics of Tobacco Opinion Leaders and Their Followers: Twitter Analysis. J Med Internet Res 2019;21(6):e12676. doi: 10.2196/12676 [published Online First: 2019/06/06]
- Cavazos-Rehg P, Li X, Kasson E, et al. Exploring How Social Media Exposure and Interactions Are Associated With ENDS and Tobacco Use in Adolescents From the PATH Study. Nicotine & Tobacco Research 2021;23(3):487-94. doi: 10.1093/ntr/ntaa113
- Albert Bandura, Walters RH. Social learning theory: Prentice Hall: Englewood cliffs. 1977.
- De Veirman M, Hudders L, Nelson MR. What Is Influencer Marketing and How Does It Target Children? A Review and Direction for Future Research. Frontiers in Psychology 2019;10 doi: 10.3389/fpsyg.2019.02685
- Forbes. Top 10 most downloaded Apps and games of 2021: TikTok, Telegram big winners 2021 [Available from: https://www.forbes.com/sites/johnkoetsier/2021/12/27/top-10-most-downloaded-apps-and-games-of-2021-tiktok-telegram-big-winners/?sh=38e68673a1fe accessed May 16 2022.
- Raymond Zhong, Frenkel S. A third of TikTok’s U.S. users may be 14 or under, raising safety questions 2020 [Available from: https://www.nytimes.com/2020/08/14/technology/tiktok-underage-users-ftc.html accessed May 16 2022.
- Rocha Á, Reis JL, Peter MK, et al., eds. The Impact of TikTok on Digital Marketing. Marketing and Smart Technologies; 2021 2021//; Singapore. Springer Singapore.
- Martínez-López FJ, López López D, eds. How TikTok’s Algorithm Beats Facebook & Co. for Attention Under the Theory of Escapism: A Network Sample Analysis of Austrian, German and Swiss Users. Advances in Digital Marketing and eCommerce; 2021 2021//; Cham. Springer International Publishing.
- TikTok. TikTok Advertising Policies - Ad Creatives & Landing Page [Available from: https://ads.tiktok.com/help/article?aid=9552 accessed July 3 2021.
- TikTok. Community Guidelines [Available from: https://www.tiktok.com/community-guidelines?lang=en accessed July 3 2021.
- Jackler RK, Li VY, Cardiff RAL, et al. Promotion of tobacco products on Facebook: policy versus practice. Tobacco Control 2019;28(1):67. doi: 10.1136/tobaccocontrol-2017-054175
- Federal Trade Commission. Disclosures 101: New FTC resources for social media influencers 2019 [updated Nov 4, 2019. Available from: https://www.ftc.gov/news-events/blogs/business-blog/2019/11/disclosures-101-new-ftc-resources-social-media-influencers accessed July 3 2021.

Reviewer 2 Report
This study adds to a growing field of tobacco marketing surveillance on social media. The authors compare the features of posts among users deemed “influencers” for large cigars and Swisher Sweets (SS). The authors have reviewed a large number of videos which will inform understanding of content youth and young adults may be exposed to. However, there are some areas for improvement.
- The rationale for focusing on large cigars and SS is unclear. Large cigar is not defined, and it is unclear how/whether the authors determine a video contained “large cigars” rather than another cigar type or the brand swisher sweets. Similarly, there is no provided rationale for only selecting SS. SS, while a popular brand, is only one brand of LCC; other social media studies indicate other brands are more prominent on social media.
- Because part of the premise is comparing videos among influencers vs noninfluencers, the definition of influencer is critical. However, the selection of 75th percentile is not supported in the text. It’s unclear why this was selected instead of 90th percentile or 95th? The authors might consider how shifting this higher or lower could impact results.
- Coding for gender/race based on appearance is problematic. I encourage the authors to review and revise this section with a lens that considers the implications of this approach. For example, the language included that coders used “all video and audio clues” implies that coders may have coded race based on voice, which may come across as linguistic profiling. While I understand the purpose is to describe potential targeting by the tobacco industry, one approach may be to reframe this to be more about the visual perception of viewers rather than coding the demographics of people.
Abstract
- Lines 22-36: There are numerous dashes throughout the abstract, perhaps a formatting error?
- Lines 28, 29: The abstract included a few lines stating “Swisher Sweets videos were more likely…” Please clarify the comparison group (more likely than large cigar videos?).
- Line 33: If possible, it would be helpful to specify or provide detail on the “differences” noted that present the urgent need for regulation. This also suggests that regulation would only apply to some products, which might be difficult in practice.
Introduction
- General: More context and detail on TikTok, including why it was selected despite not being the most commonly used social media site would be helpful. What is unique about TikTok, and why is it important to study? I was left with questions such as is this the first time TikTok has been used? What other social media platforms have been used? Also, TikTok is mentioned as the “second most popular social media platform” (line 45). As a reader, I was left wondering what the most used social media platform is and why this research was being conducted on that social media site. Add information on what precisely makes TikTok special and a relevant platform to review. Beginning on line 58, there is a section regarding payments on TikTok. It was difficult to follow and seemed unrelated to the overall analysis being completed.
- General: The citations appear outdated throughout the introduction. For example, the citations in lines 41-45 are based on 2015 data. Again, on Line 66, the cited study uses 2013 data. More recent data are available that should be included.
- Lines 49-53: The citations appear incorrect (e.g., preference for small pack size is not a finding in the Delnevo study focused on changes in the market; reduced health risk perceptions of cigar smoking compared to cigarette smoking includes a citation for a study using sales data). Suggest looking up studies on pack size, risk perceptions, and other features rather than citing studies that do not directly assess these factors.
- Line 58: The authors state the enforcement actions of the TikTok polices warrants further examination, but it is unclear why. It would be helpful to provide a reason for why the enforcement warrants further examination, particularly since this content is prohibited. For example, have other studies indicated the guidelines are not followed?
- Line 67: Please define LCCs.
- Lines 71-72: It would be helpful to provide a stronger rationale for the study goal than it is “largely unexplored.” Why is this study needed? What are the potential implications for this work? What happens if you find differences for large cigars and SS? Would the findings indicate there need to be different regulations by brand?
Methods
- Please provide additional detail on how the selected hashtags were chosen. Given the millions of hashtags, was there any attempt to delve into more niche communities in the tobacco realm? There is literature that reviews popular hashtag usage on other platforms which may be helpful to review.
- What processes were used to confirm all videos in the “large cigar” group were actual large cigars and not other products? Was there any overlap of videos in the large cigar and SS groups? Also, when did this occur – before randomization? Could those details be added to Figure 1?
- Lines 85, 91: Please clarify whether the search extended or replaced with the term SS? Line 85 states the search was extended with SS hashtags; Line 91 states the search used SS related hashtags.
- Line 94: Please provide additional details on the scraping tool and what this process entailed.
- Line 126: This is the first mention of cannabis use – it might be helpful to add details about patterns of use in the Introduction so the reader has better context for this.
- Line 130: Please clarify the meaning of “cigar-related background music” and what this has to do with making smoking gestures (theme 11).
- Lines 153-155: Were coders instructed to select unknown if items difficult to determine or if visual/audio unavailable? Or both? The text is unclear about what the instructions were.
- Lines 198-199: Please clarify the FTC recommendations and when they went into place. Specifically, were they in place prior to data collection?
Figures & Tables
- Figure 2 could be moved to supplemental or edited to provide additional images; the images presented lead to more questions and do not cover the more complex themes.
- Table 1: There are more than two categories of gender; “male” and “female” refer to sex.
- Table 1: How were the races selected?
- The descriptions in Table 1 for “product promotion” and “purchasing behavior” both include selling/purchasing. Was it possible to distinguish the seller vs buyer in all situations?
Results
- Line 274: Please clarify in the Methods/Table 1 what text is being analyzed. Is this the text within the video or the comments/description? What were the guidelines for analyzing the text?
Discussion
- General: The writing is a bit jumpy throughout. For example, at Line 328, instead of saying something was observed and moving on to the next point, it might be helpful to dig into findings on posting of minors and promoting content.
- General: It might be useful for the authors to connect these findings back to the platform features – what makes TikTok unique and why is this an important platform for this research? For example, at Lines 301-302: Isn’t a key feature of TikTok everyone using the same background music? If so, this finding doesn’t seem particularly surprising.
- Line 300: Is there a typo in this line? What does “music celebrity-endorsed LCC promotions” refer to?
- Line 319: How many videos were among retailers or cigar manufacturers?
- Line 322: The authors note that #ad and #sponsored attract attention, but it is unclear whether this is a good or a bad thing. While it may inform about the promotional nature of the posts, if it attracts attention to the post and increases exposure to tobacco content, this seems like it’d be a negative consequence?
- Line 350: How many were non-English language? These data are not included in flow chart, so, it’s unclear when this review occurred and whether it was before or after randomization?
- Line 355: Were the differences identified in the users or in the content? If the former, was there confirmation that the individuals in the videos were the users and not other people?
- Line 358: The first mention of prevention campaigns is in conclusion. I suggest moving this to the discussion and adding details on how the findings might lead to prevention.
Reviewer 3 Report
This is a very interesting and well-written paper. I have little to add, except
p.2 line 46-53. Here, I get the impression that tobacco industry promotion is the only factor associated with smoking initiation and increased smoking. Perhaps mention that social (e.g deprivation) and psychological factors (e.g sensation seeking) are relevant as well?
p.2 line 67: define LLC the first time this abbrivation is used, to inform readers who are unaware of what LCC is.
Author Response
Response to Reviewer 3 Comments
We appreciate the constructive feedback from the Reviewers. We have worked to address and integrate the feedback of each reviewer in the revised, resubmitted manuscript.
- 2 line 46-53. Here, I get the impression that tobacco industry promotion is the only factor associated with smoking initiation and increased smoking. Perhaps mention that social (e.g deprivation) and psychological factors (e.g sensation seeking) are relevant as well?
Response: Thank you for your comment and suggestion. We have included additional factors that contribute to cigar use.
Manuscript changes: In lines 45-48, we added, “Factors that contribute to the increased cigar use among youth and young adults include the availability of flavors,1 small pack sizes,2 industry’s targeted marketing,3 features that facilitate marijuana use,4 psychosocial factors5, and reduced health risk perceptions of cigar smoking compared to cigarette smoking.6”
- 2 line 67: define LCC the first time this abbreviation is used, to inform readers who are unaware of what LCC is.
Manuscript changes: We define LCCs at first mention on line 59.
References
- Sterling KL, Fryer CS, Nix M, et al. Appeal and Impact of Characterizing Flavors on Young Adult Small Cigar Use. Tobacco regulatory science 2015;1:42-53. doi: 10.18001/TRS.1.1.5 [published Online First: 2015/03/01]
- Giovenco DP, Spillane TE, Talbot E, et al. Packaging Characteristics of Top-Selling Cigars in the United States, 2018. Nicotine & Tobacco Research 2022:ntac070. doi: 10.1093/ntr/ntac070
- Ganz O, Teplitskaya L, Cantrell J, et al. Direct-to-Consumer Marketing of Cigar Products in the United States. Nicotine Tob Res 2016;18(5):864-8. doi: 10.1093/ntr/ntv190 [published Online First: 2015/09/18]
- Giovenco DP, Miller Lo EJ, Lewis MJ, et al. "They're Pretty Much Made for Blunts": Product Features That Facilitate Marijuana Use Among Young Adult Cigarillo Users in the United States. Nicotine Tob Res 2017;19(11):1359-64. doi: 10.1093/ntr/ntw182 [published Online First: 2016/09/11]
- Sterling K, Berg CJ, Thomas AN, et al. Factors associated with small cigar use among college students. American journal of health behavior 2013;37(3):325-33. doi: 10.5993/AJHB.37.3.5
- Sterling KL, Fryer CS, Fagan P. The Most Natural Tobacco Used: A Qualitative Investigation of Young Adult Smokers' Risk Perceptions of Flavored Little Cigars and Cigarillos. Nicotine Tob Res 2016;18(5):827-33. doi: 10.1093/ntr/ntv151 [published Online First: 2015/07/16]

Round 2
Reviewer 1 Report
The manuscript has a high potential for publication in "Games". The document is well structured, well developed and presents a theoretical and practical contribution to the journal. However, we suggest that the literature review be substantially improved (i.e. it is necessary to have more recent, more current articles, use scientific references from the last 5 years [Scopus and/or WoS] to make the theoretical framework more solid). We suggest that the limitations of the study and the next lines of investigation be improved. Authors should improve the introduction of the manuscript and should slightly revise the English in the final version. The study is very interesting and very pertinent. We suggest improvements to the document in the final round.
Reviewer 2 Report
The revised manuscript provides additional clarity and addresses most of the issues we identified in the first review. There are a few additional points that would be helpful to readers:
1. Lines 269-273: While Swisher Sweets is most well-known as a cigarillo brand, they are also one of the most prominent brands for large cigars. It seems clarification is needed that large cigars may have included Swisher Sweets but the differentiation was based on the branding (e.g., whether SS branding was present)? If the authors are confident that none of the cigars were Swisher Sweets, please provide additional detail on the process used to determine cigar type.
2. The terms marijuana and cannabis are used interchangeably throughout. Suggest using cannabis throughout.
3. Please update the flowchart to accurately represent the final sample of videos analyzed. As is, it appears to represent those screened but does not account for exclusions such as language.
4. Suggest revising the abstract conclusion to keep within the scope of the analyses conducted. This study highlights the prevalence of flavors/packaging characteristics in consumer-driven media. You are then making the connection that this is an industry driven tactic to garner appeal. While plenty of literature would suggest the importance of these factors for appeal, this study did not derive those findings. To say the findings support plain packaging/flavor bans is an overstatement of what this study can appropriately inform.
5. The authors state the selection of the 75th percentile was arbitrary. It seems the authors could have conducted sensitivity tests to examine what selecting 50th or 90th or other cutoffs would have generated but offer no rationale for selecting 75th. That the selection was arbitrary should be explicitly stated in the limitations to inform others who may be interested in identifying a cutoff.
